# Potential Therapeutic Application of Estrogen in Gender Disparity of Nonalcoholic Fatty Liver Disease/Nonalcoholic Steatohepatitis

**DOI:** 10.3390/cells8101259

**Published:** 2019-10-15

**Authors:** Chanbin Lee, Jieun Kim, Youngmi Jung

**Affiliations:** 1Department of Integrated Biological Science, Pusan National University, 63–2, Pusandaehak-ro, Geumjeong-gu, Pusan 46241, Korea; lcb102@pusan.ac.kr (C.L.); jieun@pusan.ac.kr (J.K.); 2Department of Biological Sciences, Pusan National University, 63–2, Pusandaehak-ro, Geumjeong-gu, Pusan 46241, Korea

**Keywords:** nonalcoholic fatty liver disease, nonalcoholic steatohepatitis, gender disparity in liver response, estrogen, liver inflammation, liver fibrosis, gender-based therapy

## Abstract

Nonalcoholic fatty liver disease (NAFLD) caused by fat accumulation in the liver is globally the most common cause of chronic liver disease. Simple steatosis can progress to nonalcoholic steatohepatitis (NASH), a more severe form of NAFLD. The most potent driver for NASH is hepatocyte death induced by lipotoxicity, which triggers inflammation and fibrosis, leading to cirrhosis and/or liver cancer. Despite the significant burden of NAFLD, there is no therapy for NAFLD/NASH. Accumulating evidence indicates gender-related NAFLD progression. A higher incidence of NAFLD is found in men and postmenopausal women than premenopausal women, and the experimental results, showing protective actions of estradiol in liver diseases, suggest that estrogen, as the main female hormone, is associated with the progression of NAFLD/NASH. However, the mechanism explaining the functions of estrogen in NAFLD remains unclear because of the lack of reliable animal models for NASH, the imbalance between the sexes in animal experiments, and subsequent insufficient results. Herein, we reviewed the pathogenesis of NAFLD/NASH focused on gender and proposed a feasible association of estradiol with NAFLD/NASH based on the findings reported thus far. This review would help to expand our knowledge of the gender differences in NAFLD and for developing gender-based treatment strategies for NAFLD/NASH.

## 1. Introduction

Nonalcoholic fatty liver disease (NAFLD), the excessive accumulation of fat in the liver that is unrelated to alcohol abuse, is a leading cause of chronic liver disease worldwide [1,2,3]. The global prevalence of NAFLD is currently estimated to be approximately 25%, which affects both adults and children [4]. Although the epidemiology and demographic characteristics of NAFLD vary worldwide, the increasing prevalence of NAFLD usually parallels the overall trend of increasing obesity and metabolic syndromes, such as type 2 diabetes and hypertension [5,6,7]. NAFLD is a spectrum ranging from nonalcoholic fatty liver (NAFL), which includes hepatic steatosis without substantial inflammation or hepatocellular injury, to nonalcoholic steatohepatitis (NASH), which is characterized by steatosis, inflammation, and hepatocellular injury with or without fibrosis [2]. Both can progress to liver cirrhosis, end-stage liver disease, or liver cancer, but the risk of progression is much greater in patients with NASH compared to the risk in those with NAFL [8,9]. Since the onset of NAFLD is usually related to unhealthy lifestyles, particularly overnutrition and/or sedentary behavior, lifestyle modification, including weight loss, dietary changes, and exercise, is required for the general management of patients with NAFL [10,11,12]. However, despite the high prevalence of NAFLD, there is currently no effective treatment for NAFLD/NASH except for liver transplantation [13]. NASH is now considered as one of the most common indications for liver transplantation and is still growing [14,15]. Hence, efforts to develop therapeutics for NAFLD/NASH are urgently required.

Estradiol (also called as 17β-estradiol) is the most abundant form of circulating estrogen [16]. It is predominantly synthesized and secreted by the ovaries in premenopausal women [17]. Hence, its level sharply declines in women during menopause, although some estradiol is also produced in other tissues, including adipose, brain, and bone tissues [16,18,19]. As the primary female hormone, estrogen mainly influences the female reproductive tract in its development, maturation, and function [20]. However, loss of estradiol, due to either natural or surgical menopause, has effects that go beyond reproductive health. Accumulating evidence has reported that menopause seems to be associated with an increased risk of cardiovascular disease, hypertension, and osteoporosis because their prevalence in postmenopausal women is higher than that in premenopausal women [21,22]. In addition, women after menopause have an increased risk of glucose intolerance, insulin resistance, hyperlipidemia, and visceral fat accumulation [23,24]. All of these symptoms are related to NAFLD [5,25,26,27]. Several population-based studies have consistently presented that the prevalence of NAFL is higher in men than women during their reproductive age, while the prevalence of NAFL among women after menopause (or after 50 years) exceeds that of men [28,29,30,31,32]. Women with Turner’s syndrome lacking endogenous estrogen production are reported to have a higher risk of NAFLD than healthy women [33]. In addition, recent studies have shown that the degree of liver fibrosis and/or the rate of progression to hepatocellular carcinoma (HCC) in NAFLD patients is higher in postmenopausal women than in premenopausal women and men [34]. These findings suggest that gender disparity in NAFLD is likely due to differences in sex hormones and that women are more resistant to the progression of NAFLD than men because of estrogen. However, the protective action of estrogen in the liver has been suggested based on epidemiological data. Thus, it remains unknown how estrogen is related to NAFLD progression. In this review, we summarized the general pathogenesis of NAFLD/NASH and the possible association of estrogen with NAFLD by reviewing the most up-to-date papers reporting the roles of sex and estrogen in NAFLD. Therefore, this review would help broaden our understanding of sex differences in the progression of NAFLD and contribute to the development of gender-based treatment strategies for NAFLD/NASH.

## 2. Pathogenesis of NAFLD/NASH

NAFLD is characterized by an excessive accumulation of fat in the liver and occurs in the absence of heavy alcohol consumption, as indicated by its name [35]. The spectrum of NAFLD ranges from NAFL to NASH and eventually to cirrhosis and/or liver cancer. NAFL (simple steatosis) refers to a condition where there is an accumulation of excess fat in the liver, which is very common and not much serious compared with NASH [36,37,38]. Although triglyceride (TG), which is a major type of lipid storage in the liver, is not normal, it rarely injures the liver by itself [39]. NASH is the more serious stage of NAFLD and involves a combination of excess fat accumulation, massive death of hepatocytes, and inflammation with or without liver fibrosis. People with NAFL rarely develop cirrhosis or liver cancer, whereas patients with NASH have a high risk for cirrhosis and liver cancer [40,41]. The key feature that distinguishes NASH from NAFL is hepatocyte death. The death rate of hepatocytes is much higher in NASH than NAFLD and positively correlates with the severity of NASH [8,9]. Lipotoxicity, which is a major cause of hepatocyte death, results from the accumulation of toxic lipid metabolites in hepatocytes because of an imbalance between hepatic lipid storage and disposal [42,43]. Hepatic fat accumulation is mainly induced by two methods of impaired lipid metabolism: the increased influx of lipids into the liver and the decreased lipid disposal in the liver [44,45,46]. Hypercaloric diets place the body in a state of energy surplus and induce the release of fatty acids from adipose depots into the bloodstream [47,48,49]. In response to these high-fat (HF) and/or high-sugar diets, a large amount of insulin released from the pancreas promotes the storage of glucose in the liver, fat, and muscles, but the liver becomes resistant to insulin, eventually leading to gluconeogenesis remaining high [50,51]. Insulin resistance and hyperinsulinemia also enhance lipolysis in adipose tissue [52,53]. As a result, a high level of plasma free fatty acids (FFAs) elevates their influx into the liver and accumulation in the liver [54,55,56]. Excessive lipid uptake leads to exaggerated lipid storage in the liver. Patients with NAFLD show an increased influx of FFAs into and de novo lipogenesis in the liver [47,57]. Enhanced de novo lipogenesis contributes to the deposition of FFAs within the liver. In normal liver, excess FFAs are largely removed through β-oxidation or export by very-low-density lipoprotein (VLDL) [58,59,60]. However, dysregulated processes of lipid removal, such as β-oxidation and/or VLDL secretion, contribute to the excessive accumulation of lipids in the fatty liver [61,62].

The accumulation of excess fats in the liver results in oxidative stress and/or endoplasmic reticulum stress and eventually leads to cellular dysfunction and apoptosis [63,64,65]. The damaged hepatocytes by lipotoxicity undergo apoptosis, and these dying hepatocytes produce reactive oxygen species (ROS), damage-associated molecular patterns, sonic hedgehog ligands, etc. [66,67,68,69]. These releasing factors further stimulate inflammatory responses by the activation of the resident macrophages and trigger fibrotic repair in the liver [70,71,72]. Because hepatocytes are the main hepatic parenchymal cells performing the major functions of the liver, severe loss of hepatocytes and subsequent compensatory proliferation of nonparenchymal cells, such as Kupffer cells, hepatic stellate cells (HSCs), and progenitor cells, result in the loss of liver function and liver failure, and eventually loss of life [73,74,75]. Therefore, the prevention of hepatocyte death by regulating lipid metabolism may be a therapeutic strategy for NASH treatment.

## 3. Estrogen and Estrogen Signaling in the Liver

Estrogens, along with progesterone, are the most important hormones for women, and four forms, namely, estrone, estradiol, estriol, and estretrol, of endogenous estrogens are produced [76]. Estrone is a less potent estrogen and is made via androstenedione conversion in adipose tissue and can be reversibly converted into estradiol in the breast and liver [77,78,79,80]. Estriol is the weakest binding ligand to the estrogen receptor and is produced by 16α-hydroxylation of estrone [81]. Estriol is present at a low level in the blood because it is rapidly excreted by urine [82]. Estretrol is detected only during pregnancy, and its physiological role is unknown [83]. Estradiol is the predominant circulating estrogen and has the highest affinity for estrogen receptor compared with other estrogens [19,84]. The concentration of circulating estradiol in the bloodstream varies from 20–1000 μg/day, depending on the menstrual cycle in premenopausal women [85]. Estrogens bind and activate two nuclear receptor isotypes, estrogen receptor α (ERα) and estrogen receptor β (ERβ), which share structural homology and ligand binding properties [86]. Both ERα and ERβ are expressed in many tissues, including the uterus, ovaries, breast, and liver, but their expression levels differ among cell types [87,88,89]. For example, ERβ is more abundant than ERα in breast cancer cells, whereas ERα is more abundant than ERβ in hepatocytes [90,91,92]. Binding of estrogen with ERs in the cytoplasm induces homo- or heterodimerization of ERs, and the complex of estrogen and ERs translocates to the nucleus where it binds to specific DNA sequences, known as estrogen response elements (EREs), and regulates the transcription of target genes, such as ATP binding cassette subfamily A member 3 (ABCA3) and growth regulation by estrogen in breast cancer 1 (GREB1) [93,94,95] (Figure 1A). In addition, estrogens trigger a variety of second messenger signaling pathways. Estrogens rapidly increase the level of cyclic adenosine monophosphate (cAMP) in breast cancer cells or intestinal epithelial cells to mediate intracellular signaling transduction, such as the mitogen-activated protein kinase (MAPK) and phosphoinositide 3-kinase (PI3K) pathways [96,97,98]. Recently, a novel estrogen receptor, that is, G-protein coupled estrogen receptor (GPER, also known as G protein-coupled receptor 30), has been reported to be located at the plasma membrane and mediate the noncanonical activity of estrogen [99]. Kuroki et al. demonstrated that estradiol bound with GPER elevated the activity of extracellular signal-regulated kinase (Erk) in the rat hippocampus [100]. Filardo et al. also showed that estradiol induced the upregulation of Erk-1 and Erk-2 in breast carcinoma cells that express neither ERα nor ERβ [101]. These studies indicate that estrogens also trigger noncanonical pathway activation via GPER (Figure 1B). However, evidence for the noncanonical pathway of estrogen is not sufficient, and further studies are needed to prove this.

Estradiol is the predominant female hormone that is necessary for the development and function of the female reproductive tract and also impacts nonreproductive organs, such as the liver [20]. The liver is not only a target tissue in which estrogen works, but it also synthesizes estradiol by converting estrone to estradiol [102]. Although the action of endogenous estradiol in the liver is still largely unknown, exogenous estradiol function in the liver has been reported in several studies. A previous report has shown that treatment with synthetic estradiol accelerates the proliferation of hepatocytes in partial hepatectomized livers of rats [103]. Galmés-Pascual et al. reported that estradiol treatment enhanced mitochondrial biogenesis and metabolic capacity in HepG2 cells, a human hepatocellular carcinoma cell line, and upregulated mitochondrial DNA and metabolism-related genes in ovariectomized female rats [104]. In addition, a recent study showed that estradiol supplementation increased liver growth and hepatocyte proliferation via GPER in zebrafish [105]. The liver has both nuclear ERs. ERα and ERβ are primarily expressed in hepatocytes and activated HSCs, respectively [92,106,107]. Because HSCs are inactivated in healthy liver, ERα is abundant, and ERβ is rarely detected or is present at very low levels in the normal liver [106,108]. The expression of ERβ is elevated in the fibrotic liver in which HSCs are activated [89,108]. However, Zhou et al. showed that both ERα and ERβ were expressed in hepatocytes of the normal liver [89]. GPER is also known to be expressed in the human liver but is undetected in the livers of mice and rats [109,110]. It remains unclear what types of cells express GPER in the human liver. A few studies have reported that HSCs, Kupffer cells, or hepatocytes express GPER [105,111,112]. Thus, identifying ER expression levels and types of ER-expressing cells is first needed to understand the role of estradiol in the liver.

The forkhead box A1 protein (FOXA1) influences the activity of estrogen/ERα complex by inducing a nucleosomal rearrangement, which allows estrogen/ERα complex to easily assess promoter-proximal regions of target genes [113,114,115,116]. In the liver, FOXA1 is known to be involved in blocking lipid accumulation in hepatocytes by inhibiting TG synthesis and promoting β-oxidation [117]. A study showed that the expression of FOXA1 in the healthy liver was higher in females than males, and its level declined in NAFL compared with healthy liver [117]. These results indicate that FOXA1 might be associated with gender disparity in NAFLD development. Li et al. demonstrated that FOXA1/2 played a dominant role in the gender dimorphism of HCC development by presenting the protective action of FOXA1/2 in female mice from diethylnitrosamine (DEN)-induced HCC development [113]. The genome-wide analysis showed that target genes of ERα are largely overlapping with those of FOXA1/2, and interaction between FOXA1/2 and ERα is important in modulating the expression of genes related to HCC resistance. However, there is no direct evidence linking FOXA1 and estrogens in the pathogenesis of NAFLD/NASH, and further studies are needed.

## 4. Gender Differences in NAFLD/NASH

The pathogenesis of NAFLD includes progressive stages from fat accumulation, to hepatocyte death, to inflammation and fibrosis [37,54]. Lipotoxicity-induced hepatocyte death is the main hallmark of the progression of NAFL into NASH [38]. It is also associated with an inflammatory and fibrotic response [66,69]. Given the higher prevalence of NAFLD in men than women and its increased risk in women at menopause and post-menopause [28,29], estradiol seems to protect the liver from hepatic injuries by suppressing lipid accumulation, inflammation, and fibrosis. Hence, it is necessary to understand the action of estrogens in the liver response to injury to reveal the sex difference in NAFLD/NASH and develop a gender-based therapeutic strategy. In this section, we summarized and discussed previous reports regarding the effects of gender, reproductive state, and estradiol on the progression of NAFLD/NASH.

### 4.1. Lipid Accumulation and Lipotoxicity

When fatty acids are either supplied in excess or their disposal is impaired, they may serve as substrates for the generation of lipotoxic species that provoke endoplasmic reticulum stress and hepatocellular injury [67]. Thus, disrupted liver metabolism leads to excessive accumulation of toxic lipids in hepatocytes. Lipotoxicity is one of the major causes of hepatocyte dysfunction and subsequent hepatocellular death, leading to the progression of NASH [9,68,118]. Accumulating evidence suggests that massive hepatocyte death is a crucial pathogenic driver that contributes to liver inflammation and fibrogenesis during NASH progression [8,9]. Upon lipotoxicity, damaged or dying hepatocytes release a variety of cytokines that activate inflammatory cells and HSCs [8,69]. Hence, lipotoxicity and subsequent hepatocellular death are considered an important link between NAFL and NASH [119]. Therefore, it is necessary to elucidate sex differences in lipotoxicity-induced hepatocellular damage and the effects of estradiol on hepatocyte survival to understand the gender differences in NAFLD/NASH.

The association of fat accumulation with estradiol in the liver has been reported in many studies [120,121,122,123,124]. Epidemiological studies presenting an elevated risk for NAFLD in postmenopausal women compared with premenopausal women suggest that estradiol protects the liver from fat accumulation [29]. This concept is also supported by animal experimental studies employing estrogen-deficient female rodents caused by ovariectomy or a knockout system [120,121,122,123,124]. Estrogen-deficient mice have disrupted lipid metabolism, such as decreased fatty acid oxidation and increased de novo lipogenesis, and showed increased lipid accumulation in the liver compared with that of normal mice [120,121]. However, supplementation with estradiol improved lipid homeostasis in the liver of estrogen-deficient mice. Estradiol supplementation also promoted the upregulation of carnitine palmitoyltransferase 1 (CPT1), which transports fatty acids into mitochondria for β-oxidation, and restored fatty acid oxidation in the livers of ovariectomized female mice [122]. Aromatase-knockout female mice that do not synthesize endogenous estrogen exhibited hepatic steatosis with downregulated β-oxidation compared with wild-type female mice [123,124]. The administration of estradiol significantly improved β-oxidation in these mice [124]. These results support the hypothesis that estrogen enhances fatty acid oxidation in the liver. In addition, Pighon et al. demonstrated that estradiol treatment reduced the levels of lipogenic genes, such as sterol regulatory element-binding protein-1c (SREBP-1c), acetyl-CoA carboxylase 1 (ACC1), and sterol-CoA desaturase 1 (SCD1), and suppressed hepatic fat accumulation in ovariectomized female rats, suggesting that estradiol is involved in hepatic lipogenesis [125]. In line with these findings, estradiol treatment in female ob/ob mice exhibiting enhanced hepatic fat accumulation downregulated the expression of lipogenesis-related genes, including fatty acid synthase (FAS), ACC1, and SCD1 [126]. A reduction in the export of lipids from the liver in the form of VLDL also elevates lipid levels in the liver. Magkos et al. presented that the rate of VLDL-TG secretion was lower in postmenopausal women and men compared with premenopausal women, indicating that estradiol influences VLDL secretion [127].

However, it remains unclear how estradiol elevates β-oxidation and VLDL secretion and alleviates lipogenesis in the liver. A few studies have shown that the hepatic ERα pathway is involved in estradiol-mediated lipid metabolism by reporting elevated fat accumulation in liver-specific ERα-knockout mice after being fed an HF diet [120]. Another study has reported that GPER is associated with sex differences in NAFL [128]. Female wild-type mice fed an HF diet were more resistant to hepatic steatosis than male wild-type mice, whereas female GPER-knockout mice were more susceptible to hepatic steatosis than male GPER-knockout mice [128]. These results suggest that the hepatoprotective effect of estradiol is mediated via GPER in the liver. However, concrete data regarding the relationship of estradiol with GPER or the ERα pathway and VLDL secretion are lacking, and further study is necessary to reveal the detailed mechanism underlying estradiol actions in the liver. In addition, no in-depth report has shown sex differences in the degree of ballooned hepatocytes or hepatocyte death in NASH, although the presence of ballooned hepatocytes is the main hallmark of NASH and an insufficient prognostic factor for advanced liver fibrosis. Therefore, it is necessary to investigate whether gender or reproductive state also affects hepatocyte death.

### 4.2. Hepatic Inflammation

Inflammation is intricately connected with hepatocellular death [129]. Damaged hepatocytes caused by lipid accumulation release paracrine stimuli to evoke hepatic inflammation, which accompanies fibrosis, eventually accelerating the transition from NAFL to NASH [65,66,72,130]. Thus, inflammation with fibrosis is the main hallmark of NASH [131]. Kupffer cells that are well known to be involved in regulating hepatic inflammation expand rapidly in the early stages of NAFLD by secreting cytokines and chemokines, such as interleukin-1β (IL-1β), interleukin-6 (IL-6), tumor necrosis factor-α (TNF-α), monocyte chemoattractant protein-1 (MCP-1), and chemokine (C-C motif) ligand 5 (CCL5), to trigger a pro-inflammatory response [70,132,133,134]. Blood-derived monocytes and bone marrow-derived neutrophils also contribute to the initiation and progression of NASH by promoting the production of pro-inflammatory mediators, such as IL-6 and TNF-α [135].

IL-6 is a well-known cytokine responsible for chronic liver inflammation, such as NASH [136,137,138]. IL-6 expression is markedly enhanced in NASH compared with healthy or simple steatotic liver [139]. In addition, there is a positive correlation between the level of IL-6 and the degree of inflammation [140], suggesting that increased hepatic IL-6 plays an important role in NASH development. Interestingly, a study reported that IL-6 was upregulated in the livers of female zebrafishes at the postmenopausal stage compared with premenopausal zebrafish, although the two groups were treated with a high-caloric diet for the same duration of time [141]. A recent study demonstrated that GPER was associated with liver inflammation in DEN-induced liver injury [111]. After being injected with DEN, GPER knockout mice exhibited enhanced inflammation and immune cell infiltration compared with those of control mice, and treatment with GPER agonist reduced the expression of IL-6 in macrophages of these knockout mice, suggesting that GPER influenced the sex-specific inflammatory response by regulating IL-6 expression. In addition, in the carcinogenic liver caused by DEN, the levels of IL-6 in both serum and liver were much higher in male than female mice [142]. Estradiol supplementation downregulated the expression of hepatic IL-6 and attenuated liver inflammation in both DEN-injected male mice and female mice that received ovariectomy. Although the DEN injury model is different from the NAFLD model, these results clearly show that IL-6 is closely associated with estradiol-mediated hepatoprotective actions. Hence, IL-6 may be involved in the estradiol-related response of NAFLD, and the detailed association of IL-6 with estradiol needs to be further investigated.

Several studies suggest that estradiol has anti-inflammatory actions in experimental animal models of NAFLD. In the animal model of NASH induced by methionine and choline-deficient (MCD) diet, male mice develop more enhanced inflammatory responses with higher immune cell infiltrations than females [143]. Administration of estradiol reduces hepatic inflammation in MCD-fed male mice, indicating the inhibitory actions of estradiol on hepatic inflammation. The ovariectomy itself promotes inflammatory responses in the liver without any hepatic injuries. Ovariectomized female rodents fed normal diet show an increase in basal level of pro-inflammatory cytokines and chemokines, including TNF-α, IL-1β, IL-6, MCP-1, macrophage inflammatory protein-2 (MIP-2), and monocyte chemokine (C-C motif) receptor 2 (CCR2), in the liver, compared to sham-operated female rodents fed same diet [144,145]. Furthermore, in two animal models of NAFLD induced by either HF and high-fructose diet or HF and high-cholesterol diet, ovariectomized female mice showed enhanced proliferation and infiltration of F4/80-positive macrophages in the liver with elevated levels of pro-inflammatory cytokines, such as TNF-α, MCP-1, CCR2, interferon-γ, and nitric oxide synthase 2, compared with sham-operated female mice [146,147]. These results support that estradiol has an essential role in suppressing hepatic inflammation. However, the direct effect of estradiol on hepatic immune cells, including Kupffer cells in NAFLD/NASH, has been studied very little. Kupffer cells are liver-resident macrophages and mainly responsible for the production of inflammatory cytokines. On the contrary, actions of estradiol in immune cells have been well described in other menopause-associated diseases, including osteoporosis [148,149,150]. Although estradiol has been shown to improve inflammatory cytokine production and phagocytic capacity of Kupffer cells in the liver by increasing Akt activation in a traumatic hemorrhage-induced liver injury model, the experimental model is different from NAFLD [151]. Thus, further studies are needed to reveal the direct effects of estradiol on Kupffer cells and other immune cells, such as monocytes and neutrophils.

### 4.3. Liver Fibrosis

Liver fibrosis is a common pathological feature of chronic liver disease, including NASH [37,131,152]. It results from unregulated wound healing and is characterized by the progressive replacement of functional hepatic tissue with collagens [37]. HSCs are the main producer of the fibrotic matrix in the liver [153,154,155]. Quiescent HSCs located in the space of Disse are activated in response to liver injury and proliferate and transdifferentiate into myofibroblast-like cells, which synthesize large amounts of extracellular matrix (ECM) components, such as type I collagen, type III collagen, fibronectin, and proteoglycans, during hepatic fibrogenesis [156]. Therefore, transdifferentiation/activation of HSCs into myofibroblast-like HSCs is a key process for the development of liver fibrosis. Epidemiological studies clearly show that sex and reproductive state are related to the risk and severity of liver fibrosis in patients with NAFLD/NASH [29,157]. Before age 50, men have an increased risk of severe fibrosis compared to women, while after the age of 50, at which point most women experience menopause, the protective effect observed in women markedly declines [29]. Postmenopausal women are also reported to have a higher risk of liver fibrosis than premenopausal women [158], suggesting that estradiol is associated with liver protection from fibrosis. In line with these findings, emerging evidence from experimental animal studies has shown that ovariectomized female rodents are at a higher risk of liver fibrosis than sham-operated female rodents, and supplementation of estradiol attenuates liver fibrosis in both males and ovariectomized females [146,147,159]. Ohashi et al. reported that ovariectomized female mice fed a simultaneous high-glucose and HF diet for 12 weeks had a significant accumulation of collagen fibrils, whereas sham-operated female mice fed the same diet hardly developed liver fibrosis, suggesting the antifibrogenic function of estradiol in NAFLD/NASH [146]. However, experimental animal models of NAFLD induced by diet, such as western, HF, or MCD diets, rarely develop severe liver fibrosis or contain only a mild degree of liver fibrosis in rodents and do not exactly reflect the NAFLD/NASH pathogenesis of humans [160,161]. Therefore, the absence of experimental animal models mimicking human NASH is a major hindrance to studying the effect of estradiol in the liver.

Recently, it has been shown that estradiol influences HSC transdifferentiation/activation. In various models of liver fibrosis caused by carbon tetrachloride (CCl_4_), dimethylnitrosamine (DMN), or thioacetamide (TAA), treatment with estradiol inhibited HSC proliferation/activation and decreased collagen production [162,163,164,165]. In addition, activated HSCs incubated with estradiol exhibited less myofibroblast-like phenotypes compared with those of vehicle-treated cells [166]. Although the detailed mechanism explaining the inhibitory function of estradiol on HSC activation remains largely unknown, it is assumed that ERβ may be involved in the action of estradiol because HSCs possess functional ERβ, not ERα [108,167]. Zhang et al. showed that treatment with an ERβ-selective antagonist, 2-Phenyl-3-(4-hydroxyphenyl)-5,7-bis(trifluoromethyl)-pyrazolo[1,5-a]pyrimidine (PHTPP), blocked the antifibrotic effects of estradiol in an experimental animal model of fibrosis [108], supporting the hypothesis that estradiol inactivates HSCs by interacting with ERβ. However, the downstream signaling pathway related to the ERβ-mediated action of estradiol in regulating HSC activation is not known. A few studies report that treatment with estradiol influences the intracellular pathways of cell survival and apoptosis in HSCs. One study showed that estradiol treatment promoted apoptosis of activated HSCs by upregulating Bcl-2 and Bcl-XL, which are antiapoptotic members of the B-cell lymphoma 2 (Bcl-2) family [89]. Another study also showed that estradiol suppressed the activation of MAPK pathways, such as ERK, p38, and c-Jun N-terminal kinases (JNK), and their transcription factors activator protein-1 (AP-1) and nuclear factor κ-light-chain-enhancer of activated B cells (NF-κB) and induced the apoptosis of activated HSCs, eventually attenuating liver fibrosis [168]. Zhang et al. recently demonstrated that estradiol treatment upregulated the expression of microRNA-29 (miR-29) by suppressing NF-κB, which is a negative regulator of miR-29, and that estradiol-mediated upregulation of miR-29 attenuated liver fibrosis in CCl_4_-induced mice [169]. Although miR-29 is well known to be downregulated in fibrotic livers and to suppress HSC activation by targeting the phosphatidylinositol 3-kinase/protein kinase B (PI3K/AKT) pathway [170], it is necessary to examine the direct target of miR-29 related to estradiol, given that miRNAs have many targets. Because estradiol-bound ERβ serves as a transcription factor, we also examined whether estradiol-bound ERβ directly promotes fibrosis-related signaling or regulates gene expression to impact HSC activation.

## 5. Gender-Based Therapy for NAFLD/NASH

For several decades, many researchers and scientists have struggled with the development of treatments for NAFLD/NASH, but now there are no Food and Drug Administration (FDA)-approved drugs for NAFLD/NASH. Since the pathogenesis of NAFLD/NASH is complex and multifaceted, there is still difficulty in finding the appropriate treatment strategy. Hence, the current therapy for NAFLD, rather than direct treatment against NASH, mainly focuses on the control of metabolic features associated with NAFLD, including obesity, diabetes mellitus, hypertension, and dyslipidemia [171,172,173,174,175]. Although a vast amount of data about various candidates or targets for NAFLD/NASH therapy are pouring in worldwide, most studies have been mainly conducted in men only. Given the sex polarities in many diseases, especially NAFLD/NASH, the development and application of drugs should take into consideration that the pathogenesis of NAFLD/NASH is different between men and women. Additionally, because the incidence and severity of NAFLD are increased among older women, particularly women post-menopause [28,29,30,31,32,158], a sex-based therapeutic strategy is needed. Moreover, pharmacological treatments, such as the required dosage and duration of administration, should be applied differently depending on sex, taking into account the different physiological characteristics between them. Therefore, it is first required that the sex differences in NAFLD/NASH be understood.

As described earlier, estradiol has the potential for hepatoprotection by reducing hepatic fat accumulation as well as suppressing liver inflammation and fibrosis (Figure 2). These beneficial actions of estradiol on the progression of liver disease raise questions about whether estrogen therapy is effective at treating liver disease. Estrogen therapy has been shown to improve a number of clinical conditions associated with menopause, including osteoporosis, vasomotor symptoms, and atherosclerosis [176,177,178]. In addition, several clinical trials on patients with diabetes suggest that estrogen therapy retains a protective effect on NAFLD [179,180,181], although there are no reports of the direct therapeutic effects of estrogen on the curation of NAFLD/NASH patients. The Women’s Health Initiative investigators monitored premenopausal women who received 0.625 mg of conjugated equine estrogen and 2.5 mg medroxyprogesterone acetate per day [179]. After up to 8.5 years of follow-up, women taking estrogen showed a lower risk of diabetes than placebo-treated women after menopause. It was also shown that estrogen therapy induced a decrease in the plasma level of cholesterol in postmenopausal women with type 2 diabetes, whereas the placebo did not. In another study, estrogen therapy reduced the serum levels of IL-6, alanine aminotransferase (ALT), aspartate aminotransferase (AST), and alkaline phosphatase (ALP) in postmenopausal women with type 2 diabetes compared with levels in the placebo-treated group [181]. Since diabetes is a well-known risk factor of NAFLD/NASH [54,67], clinical trials using estrogen therapy in diabetes suggest that estrogen therapy reduces the risk of NAFLD/NASH in postmenopausal women [182]. Furthermore, estrogen therapy has become a potential therapeutic strategy in treating male NAFLD/NASH patients. Experimental animal studies have shown that supplementation with estradiol protects the liver from various hepatic injuries in male rodents [142,183,184,185,186]. However, long-term estrogen therapy produces side effects, such as headaches, nausea, and vein thrombosis, and a small risk of breast cancer in postmenopausal women [182], and these side effects are not known in men. Therefore, more studies are needed to inhibit the side effects and increase the beneficial effect of estrogen in NAFLD/NASH.

ERs are suggested as potential targets for developing NAFLD/NASH therapeutics. Chow et al. reported that the ERα-specific agonist 16α-LE2 suppressed hepatic steatosis in male mice [185]. Zhang et al. showed that the ERβ-specific agonist diarylpropionitrile (DPN) reduced the proliferation and activation of HSCs [108]. β-LGND2, an ERβ-selective agonist, has also been shown to prevent the inflammatory response and fibrosis in the liver of male mice fed an HF diet [186]. These results suggest that both ERα and ERβ agonists are one of many candidates for treating NAFLD/NASH. Recently, researchers have focused on the active metabolite of estradiol [163,187,188,189]. Estradiol is the most active estrogen in vivo and is converted to several metabolites. Some of these metabolites are known to have biological activities. 2-Hydroxyestradiol (2OHE) and 2-methoxyestradiol (2MeOE) are biologically active substances, and they have been mainly investigated. 2OHE, which is one of the hydroxylated metabolites of estradiol, is converted by cytochrome P-450 enzymes and is rapidly catalyzed to 2MeOE by catechol-O-methyl-transferase. Multiple studies have recently shown that 2OHE and 2MeOE inhibit DNA synthesis, cell proliferation, collagen synthesis, and MAPK activity in vascular smooth muscle cells, cardiac fibroblasts, and glomerular mesangial cells, and their effects are stronger than estradiol [187,188,189,190]. In addition, these metabolites have an antifibrotic function in liver fibrosis. 2OHE and 2MeOE are shown to inhibit HSC proliferation and ECM secretion in vitro [163]. Estradiol exerts antifibrotic action at only the high concentration (≥ 10^−7^ mol/L), while 2MeOH presents a similar effect at as low as 10^−9^ mol/L. Given that the metabolites of estradiol are known to have little affinity for ERs [191,192,193] and have inhibitory effects on liver fibrosis at very low concentrations compared with estradiol [163], they are strong candidates for potential medicine for NAFLD/NASH. In addition, they are good research targets for investigating and understanding how endogenous estradiol protects the liver from hepatic injuries. The liver is the main site where estradiol is metabolized [80]. Hence, the antifibrogenic actions of estradiol may be mediated in part by the local conversion of estradiol to biologically active metabolites. However, further studies are needed to investigate whether and how the metabolites of estradiol influence liver fibrosis.

## 6. Conclusions

The prevalence of NAFLD/NASH is increasing worldwide, but no therapies that primarily target NAFLD/NASH have been approved. A higher risk of NAFLD/NASH in men and postmenopausal women than premenopausal women and experimental data for the effects of estrogen in the liver indicate that estrogen is involved in NAFLD/NASH [28,29,30,31,32]. However, limited data providing the functions of estrogens in the progression of NAFLD/NASH hinder the application of estrogen in treating liver disease. In addition, the lack of animal experimental models reflecting human NASH and the unbalanced experimental design regarding the sex of the animals are hurdles to overcome in developing NASH treatment. Therefore, it is necessary to develop and design sex-balanced animal experiments that are more reproducible for human NASH. In addition, more accurate epidemiological and pathophysiological data obtained from larger cohort studies are required in order to examine and trace the gender differences in human patients with NAFLD/NASH. It would help facilitate gender-based therapy for NASH by selecting a more appropriate choice for NASH treatment. Finally, further studies are needed to investigate the detailed mechanism underlying the hepatoprotective actions of estradiol and the therapeutic safety of applying estrogen therapy for treating liver disease.

## Figures and Tables

**Figure 1 cells-08-01259-f001:**
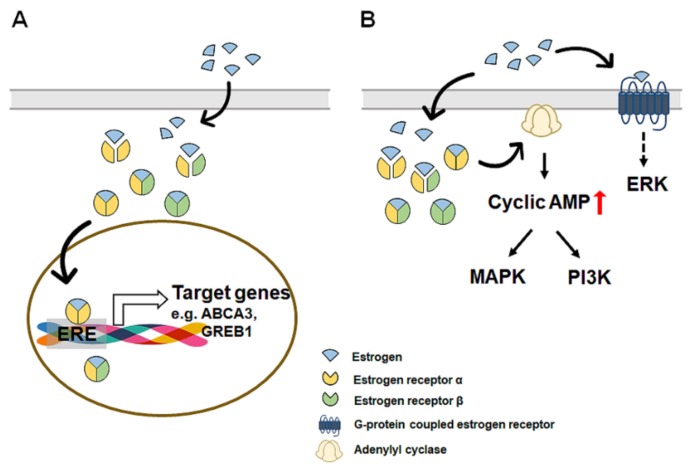
A simplified diagram of estrogen signaling pathways, including genomic (canonical) and nongenomic (non-canonical) pathway. The intercellular effects of estrogen are mediated by two main pathways: genomic (canonical) and nongenomic (non-canonical). In the genomic pathway (**A**), estrogens (blue colored) bind and activate two nuclear receptor isotypes, estrogen receptor α (ERα, colored with yellow) and estrogen receptor β (ERβ, colored with green). Binding of estrogen with ERs in the cytoplasm induces homo- or heterodimerization of ERs, and the complex of estrogen and ERs translocate to the nucleus where it binds to specific DNA sequences, known as estrogen response elements (EREs), and regulates the transcription of target genes, such as ATP binding cassette subfamily A member 3 (ABCA3) and growth regulation by estrogen in breast cancer 1 (GREB1). In the nongenomic pathway (**B**), estrogens activate several signaling pathways without direct interaction with DNA. Estrogen rapidly increases the level of cyclic adenosine monophosphate (cAMP) by activating adenylyl cyclase and mediates intracellular signaling transduction, such as the mitogen-activated protein kinase (MAPK) and phosphoinositide 3-kinase (PI3K) pathways. Estrogens also bind to G-protein coupled estrogen receptor (GPER) at the plasma membrane and mediate the activity of extracellular signal-regulated kinase (Erk).

**Figure 2 cells-08-01259-f002:**
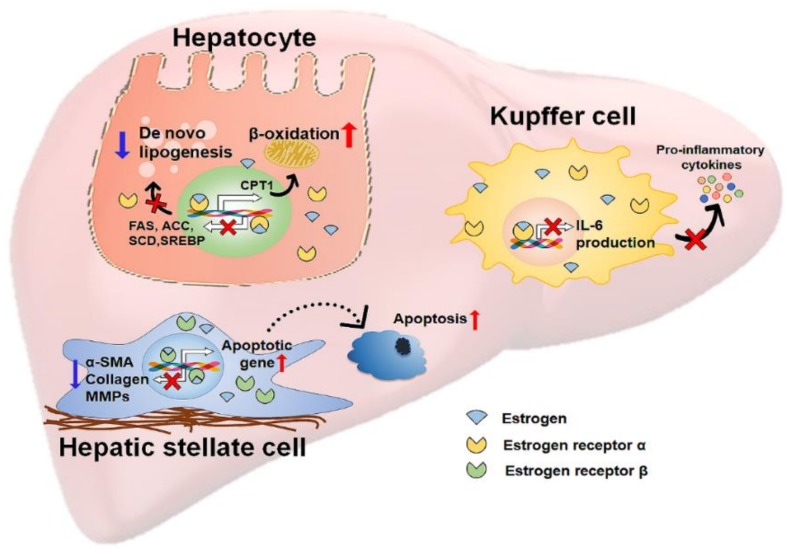
A schematic depicting the protective actions of estrogen in the liver. The scheme depicts the hepatoprotective actions of estrogen against nonalcoholic fatty liver disease (NAFLD). Estrogens bind to estrogen receptors (ERs) within hepatic cells and translocate into the nucleus of target cells where they regulate gene expression. In hepatocytes, estrogen bound with ERα (colored as yellow) alleviates lipotoxic stress in these cells by suppressing de novo lipogenesis and promoting β-oxidation. Estrogen/ERα decreases the expressions of de novo lipogenesis-related genes, such as fatty acids synthase (FAS), acetyl-CoA carboxylase (ACC), stearoyl-CoA desaturase (SCD), and sterol regulatory element-binding protein (SREBP) and increases the expressions of β-oxidation-related genes, such as carnitine palmitoyltransferase 1 (CPT1). Estrogen binds to ERα in Kupffer cells, which are liver-resident macrophages. Estrogen-bound ERα inhibits the production of IL-6 and the secretion of pro-inflammatory cytokines in these cells, eventually reducing inflammation. In HSCs, estrogens bound with ERβ upregulate apoptosis-related genes and downregulate profibrotic genes, such as α-smooth muscle actin (α-SMA), collagen, and matrix metallopeptidases (MMPs), attenuating liver fibrosis.

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
