# Peer review of "Potential Therapeutic Application of Estrogen in Gender Disparity of Nonalcoholic Fatty Liver Disease/Nonalcoholic Steatohepatitis"

_cells, 2019, doi:10.3390/cells8101259_

Round 1

Reviewer 1 Report

The review authors provide available evidence on the role of gender difference in the development of NAFLD. Evidence-based data of the article confirms the potential therapeutic application of estrogen in nonalcoholic fatty liver disease. I cannot but agree with the authors' reasoning. However, the line 89 saying that “simple steatosis is not a serious problem” seems to be disputable. The reason is that the patients with steatosis in the absence of any inflammation have a risk of developing fibrosis over 15-20 years. So, it is preferable (can be recommended) to replace the word "poor" with a synonym (21, 222, 234 line).

Author Response

Reviewer #1

The review authors provide available evidence on the role of gender difference in the development of NAFLD. Evidence-based data of the article confirms the potential therapeutic application of estrogen in nonalcoholic fatty liver disease. I cannot but agree with the authors' reasoning. However, the line 89 saying that “simple steatosis is not a serious problem” seems to be disputable. The reason is that the patients with steatosis in the absence of any inflammation have a risk of developing fibrosis over 15-20 years. So, it is preferable (can be recommended) to replace the word "poor" with a synonym (21, 222, 234 line).

: As you suggested, we changed “not serious” with “ not much serious compared with NASH” in the revised manuscript. In addition, we replaced “poor” with “ NAFLD remains unclear” in line 21, “However, it remains unclear how” in line 223, and “insufficient prognostic” in line 235.

Reviewer 2 Report

The authors nicely give an update on the current status on estrogen role in NAFLD/NASH.  The review is logically set up and the flow of the text is good.

Since the title is Gender Differences” I expected to find a discussion also on the role of androgens, which have been shown to play a role in inflammation of the liver. I suggest to add some information also on such aspect.

Sentence in line 47 "NASH is now considered the most common indication for liver transplantation" is not supported by references and seem to be too strong. I would suggest a simple “one of the most common”.

Description of the different types of estrogen in introduction is redundant since a specific paragraph is dedicated to them.

In my opinion Figure 1 are not useful for the readers. Some images on pathogenesis of NAFLD/NASH and estrogen signaling may be more interesting and impactful.

I find that the paragraph 4.2 Hepatic inflammation (check the typo), is not fully covering the issue. Indeed, discussion are limited to IL6 but also other cytokines as well as the recruitment of inflammatory cells could be modulated by sexual hormones. I would suggest that the authors elaborate a bit on this.

Author Response

Reviewer #2

The authors nicely give an update on the current status on estrogen role in NAFLD/NASH.  The review is logically set up and the flow of the text is good.

Since the title is Gender Differences” I expected to find a discussion also on the role of androgens, which have been shown to play a role in inflammation of the liver. I suggest to add some information also on such aspect.

: This review focuses on estrogen in NAFLD. If androgen is added in the manuscript, there are too many contents which we should explain. It will be too extensive. Hence, we changed the title to highlight association of estrogen in NAFLD in the revised manuscript; “Potential therapeutic application of estrogen in gender disparity of nonalcoholic fatty liver disease/nonalcoholic steatohepatitis.

Sentence in line 47 "NASH is now considered the most common indication for liver transplantation" is not supported by references and seem to be too strong. I would suggest a simple “one of the most common”.

: As you suggested, we changed it with “ ~considered as one of the most ~~~”. In addition, we added the references [J Hepatol. 2019 Aug;71(2):313-322. / Liver Transpl. 2019 Jan;25(1):10-11.] for that in the revised manuscript.

Description of the different types of estrogen in introduction is redundant since a specific paragraph is dedicated to them.

: Thank you for your comment. Repeated part in the introduction was deleted in the revised manuscript: “Estradiol (also called as 17β-estradiol) is the most abundant form of circulating estrogen. It is predominantly ”.

In my opinion Figure 1 are not useful for the readers. Some images on pathogenesis of NAFLD/NASH and estrogen signaling may be more interesting and impactful.

: As your suggestion, we replaced it with new figure 1 presenting the estrogen-related signaling pathway in the revised manuscript. Because pathogenesis of NAFLD/NASH is shown in many review and research papers, we did not add the image for that. However, we will add the images, if you think it is necessary.

I find that the paragraph 4.2 Hepatic inflammation (check the typo), is not fully covering the issue. Indeed, discussion are limited to IL6 but also other cytokines as well as the recruitment of inflammatory cells could be modulated by sexual hormones. I would suggest that the authors elaborate a bit on this.

: Thank you for your correction. As your suggestion, we added more descriptions about the role of estradiol in hepatic inflammation in section ‘4.2 Hepatic inflammation’; “Several studies suggest that estradiol has anti-inflammatory actions in experimental animal models of NAFLD. In animal model of NASH induced by methionine and choline deficient (MCD) diet, male mice develop more enhanced inflammatory responses with higher immune cell infiltrations than females [Exp Biol Med (Maywood). 2015 Oct; 240(10): 1279–1286.]. Administration of estradiol reduces hepatic inflammation in MCD-fed male mice, indicating the inhibitory actions of estradiol on hepatic inflammation. The ovariectomy itself promotes inflammatory responses in the liver without any hepatic injuries. Ovariectomized female rodents fed normal diet show an increase in basal level of pro-inflammatory cytokines and chemokines including TNF-α, IL-1β, IL-6, MCP-1, macrophage inflammatory protein-2 (MIP-2), and monocyte chemokine (C-C motif) receptor 2 (CCR2) in the liver, compared to sham-operated female rodents fed same diet [Oncotarget. 2015 May 10; 6(13): 10801–1081, Endocrinology. 2009 May; 150(5): 2161–2168]. Furthermore, in two animal models of NAFLD induced by either HF and high-fructose diet or HF and high-cholesterol diet, ovariectomized female mice showed enhanced proliferation and infiltration of F4/80-positive macrophages in the liver with elevated levels of pro-inflammatory cytokines such as TNF-α, MCP-1, CCR2, interferon-γ, and nitric oxide synthase 2, compared with sham-operated female mice [Food Chem Toxicol. 2018 Aug;118:190-197., Am J Physiol Gastrointest Liver Physiol. 2011 Dec;301(6):G1031-43]. These results support that estradiol has an essential role in suppressing hepatic inflammation. However, the direct effect of estradiol on hepatic immune cells including Kupffer cells in NAFLD/NASH have been studied very little. Kupffer cells are liver-resident macrophages and mainly responsible for production of inflammatory cytokines. On the contrary, actions of estradiol in immune cells have been well described in other menopause-associated diseases including osteoporosis [J Clin Invest. 2006 May 1; 116(5): 1186–1194. / Proc Natl Acad Sci U S A. 1991 Jun 15; 88(12):5134-8./ J Clin Invest. 2000 Nov; 106(10):1229-37.]. Although estradiol was shown to improve inflammatory cytokine production and phagocytic capacity of Kupffer cells in the liver by increasing Akt activation in traumatic hemorrhage-induced liver injury model, the experimental model is different from NAFLD. Thus, further studies are needed to reveal the direct effects of estradiol on Kupffer cells and other immune cells such as monocytes and neutrophils.”

Reviewer 3 Report

The present manuscript entitled " Gender differences imply the potential therapeutic …." described a review on the pathogenesis of
NAFLD/NASH related to the gender and the possible involvement of estrogen. This review is informative. However, some minor points should be
added to help the reader:

-In paragraph 3, authors described the role of estrogen and it signaling pathway in the liver. A short figure summarizing these aspects could be includedin this paragraph.

- The title of paragraph 4.2 is misspelled

Author Response

Reviewer #3

The present manuscript entitled " Gender differences imply the potential therapeutic …." described a review on the pathogenesis of NAFLD/NASH related to the gender and the possible involvement of estrogen. This review is informative. However, some minor points should be added to help the reader:

-In paragraph 3, authors described the role of estrogen and it signaling pathway in the liver. A short figure summarizing these aspects could be included in this paragraph.

: Because we already provided the image showing the actions of estrogen in liver (figure 2), we presented the new figure 1 which summarized estrogen-related signaling pathway in the revised manuscript.

- The title of paragraph 4.2 is misspelled

: Thank you for your correction. We corrected it in the revised manuscript.

Reviewer 4 Report

This is a very interesting review of sex differences in NAFLD. It is well written. I only have a few questions/comments.

Page 2 line 49/50: The wording here is strange. I am not really comfortable with calling E2 estrogen. While you acknowledge the others, most anyone who is familiar with endocrinology would not consider this 'estrogen'. It is the most abundant and the most studied but I would consider rewording.

Page 3 line 124/125: This is line and some that follow are almost identical to those in in the section listed above only without the names for the 4 estrogens.

Page 8: Are the estrogen receptor #s and ratios the same in male and female hepatocytes? Even if they are there is a distinct possibility that estrogen will not have the same effects in males as in females. The timing of replacing estrogen can also impact the results.

I am not sure that I agree with the use of estrogen therapy; however, I agree strongly with your actual conclusion and the discussion is evidence based. 

Author Response

Reviewer #4

This is a very interesting review of sex differences in NAFLD. It is well written. I only have a few questions/comments.

Page 2 line 49/50: The wording here is strange. I am not really comfortable with calling E2 estrogen. While you acknowledge the others, most anyone who is familiar with endocrinology would not consider this 'estrogen'. It is the most abundant and the most studied but I would consider rewording.

: We agreed with your comment. We changed estrogen with estradiol in the revised manuscript.

Page 3 line 124/125: This is line and some that follow are almost identical to those in in the section listed above only without the names for the 4 estrogens.

: Repeated part in the introduction was deleted in the revised manuscript; “Estradiol (also called as 17β-estradiol) is the most abundant form of circulating estrogen. It is predominantly ”.

Page 8: Are the estrogen receptor #s and ratios the same in male and female hepatocytes? Even if they are there is a distinct possibility that estrogen will not have the same effects in males as in females. The timing of replacing estrogen can also impact the results.

: It does not know whether the number of estrogen receptor and its efficiency are different between male and female. However, we catch your meaning and totally agree with you. The number of estrogen receptor and its efficiency are possibly different between man and woman and these differences influence the therapeutic effect of estrogen in liver. That is the reason why it is necessary to develop and design sex-balanced animal experiments that are more reproducible for human NASH, as we described in the manuscript (conclusion part). In addition, we also agree with your comments that the timing of replacing estrogen can impact the results. Therefore, we describe, more accurate epidemiological and pathophysiological data obtained from larger cohort studies are required in order to examine and trace the gender differences in human patients with NAFLD/NASH” in the conclusion section. Prior to application of estrogen in therapeutics, more in-depth studies are required to reveal the detailed mechanism underlying the hepatoprotective actions of estrogen to develop safe and optimal treatment strategies for treating liver disease. We addressed these points in the conclusion parts of the first version of manuscript.

I am not sure that I agree with the use of estrogen therapy; however, I agree strongly with your actual conclusion and the discussion is evidence based.

Reviewer 5 Report

The authors reviewed the role of estrogen and its-related signaling pathways in the gender difference  of NAFLD, NASH and liver cancer development.

However, in terms of gender disparity in human liver diseases and cancer, the followings needs to be further addressed. 

Association of estrogen receptor-FOXA1/2 signaling pathways and also its mutual relationship to androgen receptor-FOXA1/2 signaling pathways. 

Li Z., Tuteja G., Schug J., Kaestner K.H.
Foxa1 and Foxa2 are essential for sexual dimorphism in liver cancer. Cell, 148 (2012), pp. 72-83.

Ma W.L., Hsu C.L., Yeh C.C., Wu M.H., Huang C.K., Jeng L.B., et al. Hepatic androgen receptor suppresses hepatocellular carcinoma metastasis through modulation of cell migration and anoikis
Hepatology, 56 (2012), pp. 176-185

Sugathan A, Waxman DJ. Genome-wide analysis of chromatin states reveals distinct mechanisms of sex-dependent gene regulation in male and female mouse liver. Mol Cell Biol. 2013 Sep;33(18):3594-610.

Ma WL, Lai HC, Yeh S, Cai X, Chang C. Androgen receptor roles in hepatocellular carcinoma, fatty liver, cirrhosis and hepatitis. Endocr Relat Cancer. 2014 May 6;21(3):R165-82. doi: 10.1530/ERC-13-0283.

Feng H, Yu Z, Tian Y, Lee YY, Li MS, Go MY, Cheung YS, Lai PB, Chan AM, To KF, Chan HL, Sung JJ, Cheng AS. A CCRK-EZH2 epigenetic circuitry drives hepatocarcinogenesis and associates with tumor recurrence and poor survival of patients. J Hepatol. 2015 May;62(5):1100-11.

Author Response

Reviewer #5

The authors reviewed the role of estrogen and its-related signaling pathways in the gender difference of NAFLD, NASH and liver cancer development. However, in terms of gender disparity in human liver diseases and cancer, the followings needs to be further addressed.

Association of estrogen receptor-FOXA1/2 signaling pathways and also its mutual relationship to androgen receptor-FOXA1/2 signaling pathways.

: This review focuses on estrogen in NAFLD. If androgen is added in the manuscript, there are too many contents which we should explain. It will be too extensive. Hence, we added the estrogen receprot-FOXA1/2 signaling pathway in the revised manuscript; “The forkhead box A1 protein (FOXA1) influences the activity of estrogen/ERα complex by inducing a nucleosomal rearrangement, which allows estrogen/ERα complex to easily assess promoter proximal regions of target genes [Cell, 148 (2012), pp. 72-83./ Nat Genet. 2011 Jan;43(1):27-33/ Chin J Cancer. 2014 Feb;33(2):51-67/Cell. 2016 Apr 21; 165(3): 593–605]. In the liver, FOXA1 is known to be involved in blocking lipid accumulation in hepatocytes by inhibiting TG synthesis and promoting β-oxidation [PLoS One. 2012; 7(1): e30014.]. Expression of FOXA1 in healthy liver was higher in females than males, and its level declined in NAFL compared with healthy liver [PLoS One. 2012; 7(1): e30014.]. These results indicate that FOXA1 might be associated with gender disparity in NAFLD development. Li et al demonstrated that FOXA1/2 plays a dominant role in the gender dimorphism of HCC development by presenting the protective action of FOXA1/2 in female mice from DEN-induced HCC development. [Cell, 148 (2012), pp. 72-83]. Genome-wide analysis showed that target genes of ERα are largely overlapping with those of FOXA1/2, and interaction between FOXA1/2 and ERα is important in modulating expression of genes related to HCC resistance. However, there is no direct evidence linking FOXA1 and estrogens in pathogenesis of NAFLD/NASH, and further studies are needed.”

In addition, we changed the title to highlight association of estrogen in NAFLD in the revised manuscript; “Potential therapeutic application of estrogen in gender disparity of nonalcoholic fatty liver disease/nonalcoholic steatohepatitis"

Round 2

Reviewer 2 Report

The paper has been improved and I do not have additional comments for the Authors.

Reviewer 5 Report

The role of FOXA1-ER alpha signaling pathways in gender disparity was addressed.